# Practical research on the boundaries of MAYA design principles with ceramic products as the carrier

Qi Wu [1,2], Mohd Faiz bin Yahaya [1‡*], Lichen Tai [3‡], Qianhui Ren [1‡]

**1** Department of Industrial Design, Faculty of Design and Architecture, Universiti Putra Malaysia, Serdang Selangor, Malaysia, **2** Lyuliang University, Shanxi, China, **3** Wenzao Ursuline University of Languages, Taiwan, China

☯ These authors contributed equally to this work.
‡ These authors also contributed equally to this work.
\* mfaizy@upm.edu.my

## Abstract

Both typicality and novelty shape aesthetic preference—typicality as conformity to a product's prototype, and novelty as perceived difference and originality. While prior research has highlighted the tension and balance between these two factors, few studies have examined how product category structure moderates their effects. This study aims to investigate the roles of typicality and novelty in shaping consumer preferences, as well as whether product type (poor vs. rich categories) influences the relative impact of these factors. To achieve this, we conducted a within-subjects experiment using 20 ceramic product stimuli: 10 ceramic vases(poor categories) and 10 ceramic lighting items (rich categories). A total of 200 Chinese participants evaluated the typicality, novelty, and liking of each product using 7-point Likert scales. Data were analyzed using repeated-measures ANOVA, partial correlation, generalized estimating equations (GEE), and hierarchical regression. Findings revealed that typicality was more predictive in poor categories, while novelty played a greater role in rich categories. Moreover, product category structure significantly moderated these effects, confirming and refining the boundary conditions of the MAYA (Most Advanced Yet Acceptable) principle. Furthermore, regression analyses revealed that typicality and novelty together explained 21.8% of the variance in aesthetic preference for ceramic vases and 20.3% for ceramic lighting items. This research deepens theoretical understanding of aesthetic judgment by highlighting the contextual role of category structure. It also provides practical design guidance, emphasizing typical features in poor categories and prioritizing novelty in rich categories to optimize consumer appeal.

**Data availability statement:** All relevant data are within the manuscript and its Supporting Information files (S1 file, S2 file).

**Funding:** The author(s) received no specific funding for this work.

**Competing interests:** The authors have declared that no competing interests exist.

## Introduction

In the field of empirical aesthetics, aesthetics is understood as a systematic process of perceptual experience that explores how individuals derive pleasure, meaning, and value from the perception of form, color, and material [1–3]. In product design, this framework is particularly relevant for understanding how design features elicit aesthetic pleasure, which represents the immediate emotional response to perceiving an object, independent of its utilitarian function [4]. Building on this affective foundation, preference reflects emotion-driven inclinations influenced by sensory and affective factors [5]. Aesthetic preference, in turn, refers to the degree to which a stimulus is favored due to the aesthetic pleasure it elicits [6]. This preference integrates both affective and cognitive components of the aesthetic experience, which is crucial in guiding design decisions that balance novelty and user acceptance.

In contemporary markets driven by consumer experience, product visual design plays a central role in shaping purchase behavior, elevating user satisfaction, and reinforcing brand perception. Consumers' aesthetic judgments are shaped by their personal identity, cultural context, and emotional needs, offering designers valuable cues for anticipating user preferences and refining design strategies [7]. Embedding aesthetic considerations throughout the product development cycle not only enhances visual appeal but also deepens emotional engagement, improves overall user experience, and cultivates stronger brand loyalty [8,9].

Among various aesthetic media, ceramics—one of the earliest materials used by humankind—combine utility with expressive form. From traditional vessels to contemporary home décor and art installations, ceramic products have consistently occupied a unique space at the intersection of function and culture. They simultaneously fulfill everyday needs and serve as representations of individual taste and lifestyle. In modern contexts, the role of ceramics in home design, interior styling, and art consumption has grown considerably. According to Grand View Research (2024), the global ceramics market was valued at USD 248.89 billion in 2023 and is projected to reach USD 359.35 billion by 2030, growing at a CAGR of 5.6%. Wu et al. (2024) noted that both functionality and visual appeal are decisive factors in consumers' evaluations of ceramic products. Especially in highly saturated markets, aesthetic design has become a crucial point of differentiation [10].

To assess the current academic landscape of ceramic product aesthetics, this study conducted a literature search in the Web of Science database using the keywords 'ceramic product' and 'aesthetic.' The search yielded only two relevant publications, indicating that this research area remains in its early stages and offers substantial potential for scholarly exploration. For instance, Liu (2024), drawing on Kansei Engineering theory, combined eye-tracking technology with semantic differential scales to investigate users' perceptual experiences while interacting with ceramic teapots [11]. In another example, María-Jesús Agost (2014) examined the relationship between emotion, meaning, and aesthetic preference, analyzing how individual values influence judgments of ceramic tile aesthetics [12]. Despite the rich cultural symbolism and visual diversity of ceramic products, research on their aesthetic judgment remains limited. It is even less developed within the theoretical frameworks

of product categorization and aesthetic evaluation. This underrepresentation is surprising given ceramics' dual nature as both functional tools and aesthetic artifacts.

Nevertheless, the theoretical mechanisms underlying how visual design influences aesthetic judgment remain insufficiently clarified. Existing studies suggest that two primary cognitive dimensions—typicality and novelty—underlie aesthetic appraisal. Typicality refers to the extent to which a product represents or conforms to the prototype of its category [13,14]. It reflects the degree of congruence between a product and the mental representation or schema that individuals hold for a particular product category. In other words, typicality captures the similarity between a given product and the core prototype of its category [15]. According to prototype theory, individuals tend to prefer stimuli that are closer to category norms, as these facilitate easier recognition and classification [16]. Furthermore, typicality has been shown to correlate with cognitive fluency—designs closer to the prototype are more likely to be accepted [17].In contrast, Novelty refers to the degree to which a stimulus diverges from prior experience or expectation [18], thereby stimulating curiosity and perceptual arousal. It has been found to enhance product appeal [19], communicate technological advancement [20], and increase perceived uniqueness [21].

The tension between novelty and typicality poses a classic aesthetic dilemma: while typicality fosters familiarity but may lack distinctiveness, conversely, novelty generates interest, but it may also impose cognitive load. This trade-off between cognitive familiarity and perceptual stimulation has prompted researchers to seek theoretical models that reconcile these opposing forces. To reconcile this paradox, Loewy (2002) proposed the MAYA principle— 'Most Advanced Yet Acceptable'—suggesting that effective design should strike a balance between novelty and recognizability [22]. Empirical support for the MAYA principle has been demonstrated across multiple studies [23,24], indicating that a well-balanced design can maximize aesthetic pleasure and consumer preference. However, research has also shown that the optimal balance between novelty and typicality may vary by product category, raising questions about the universal applicability of the MAYA principle [25–28].

While the MAYA principle offers a compelling account of design preference, recent studies suggest that contextual factors—such as product category structure—may moderate its effects. To further define the boundary conditions of the MAYA principle, this study adopts the Categorical-Motivational (CM) model, which posits that object categories influence the relative weights of typicality and novelty in shaping aesthetic preference [29]. An essential feature of the CM model is that it is bipolar. At one extreme are categories that are largely formed and closed to further articulation, while at the other extreme are categories that are open to further articulation [30]. Closed categories permit no novel exemplars for the individual, and positive affect is primarily accounted for by prototypicality [30]. Prototypicality closely follows the category prototype and can be processed with minimal cognitive effort [31,32]. In contrast, open categories "demand" novel exemplars, and positive affect is accounted for primarily by novelty [30].

Falling between these two extremes are the partially open categories, which encompass most real-world categories [33]. In such categories, novel exemplars are permitted, and positive affect is influenced by both novelty and prototypicality [30]. It is argued that the foundation of novelty lies in categorization and the information stored in the brain's individual object representations, with variation in these representations ranging from 'rich' to 'poor' [34]. Within this framework, partially open categories can be further differentiated into poor and rich categories.

Based on Rosch's hierarchical categorization (superordinate, basic, and subordinate categories) [35], the classification of poor and rich categories begins at the basic level [36]. For example, chairs belong to the basic level category, from which subordinate categories are formed according to the method of knowledge acquisition. Products classified as poor categories have relatively few subordinate categories, whereas products in rich categories include numerous subordinate categories—for instance, chairs comprise various subcategories such as Reclining chairs, boss chairs, dining chairs, etc. [36]. Subsequent empirical studies have supported this model [34,37–39], demonstrating that category structure systematically moderates consumer tolerance for design variation. This distinction suggests that consumers may process aesthetic attributes differently depending on the structural flexibility of the product category, thus shaping their sensitivity to novelty versus typicality.

However, existing research in this area has primarily focused on furniture, apparel, packaging, and consumer electronics. The ceramic domain—characterized by its unique cultural and formal attributes—has received limited attention in terms of product category structure. Furthermore, studies on stimuli from rich categories are often confined to 'chairs,' suggesting a need to expand the conceptual scope.

In response, this study classifies ceramic vases as products belonging to poor categories, as they are not typically subdivided into further subcategories. In contrast, ceramic lighting devices are considered a rich category, as they can be further differentiated into subtypes such as lamps, spotlights, and chandeliers [36]. This study examines the impact of typicality and novelty on aesthetic preferences and whether product category structure moderates this relationship, thereby refining the applicability of the MAYA principle. Previous research has shown that poor categories tend to favor high typicality and low novelty [33,37–39]. In contrast, rich categories tend to favor low typicality and high novelty [33]. However, this pattern has not yet been tested in the context of ceramic products. Therefore, the present study proposes the following core hypothesis, which serves as the central organizing principle of the research: For ceramic products, rich categories are more tolerant of novelty than poor categories. All subsequent analyses are designed to test this hypothesis by examining how typicality and novelty differentially predict aesthetic preference across category structures. This work extends aesthetic theory into culturally expressive product domains and introduces ceramics as a testbed for category-based design research. Drawing upon the aesthetic dimensions of typicality, novelty, and aesthetic preferences [4], this study seeks to: (1) examine the relative influence of typicality and novelty on aesthetic preference; (2) test whether this influence differs across poor and rich categories; and (3) assess whether product category structure moderates the applicability of the MAYA principle.

## Materials and methods

### Stimuli

In the present study, stimuli were carefully selected from the market to ensure both representativeness and diversity, spanning a broad range of typicality and novelty. For each stimulus category, 50 candidate images were initially collected, resulting in a total of 100 images (50 ceramic vases and 50 ceramic lighting items) from the Chinese e-commerce platform Taobao. The expert panel consisted of five members from the UMA research team. These five domain experts independently evaluated all candidate images along two dimensions—typicality and novelty—using a three-point scale (1 = high typicality, 2 = moderate typicality/novelty, 3 = high novelty). Based on these ratings, 10 images were selected from each category, including 3 with high typicality, 4 with moderate typicality/novelty, and 3 with high novelty.

In the provided visual stimuli, all brand and specification information was deliberately removed to ensure the evaluation process was based solely on the product's visual design attributes. To ensure consistency across stimuli within each category, all products were presented from a frontal view, and the background color was standardized to white to avoid visual distractions. Fig 1 illustrates all the stimuli (including ceramic vases and ceramic lighting items) and their corresponding numbering or labeling methods, which simplify subsequent data analysis and thereby improve identification efficiency and data analysis accuracy. Additionally, the number of stimuli was determined in a pilot study to ensure statistically significant differences in typicality and novelty, thereby improving the reliability and validity of the experimental results.

### Inclusivity in global research

Additional information regarding the ethical, cultural, and scientific considerations specific to inclusivity in global research is included in the Supporting Information (S1 Checklist)

### Participants

Between September and October 2024, participants in this study were recruited from among teachers, students, and other staff members at Lyuliang University in China, who participated in a face-to-face product evaluation survey. To ensure an

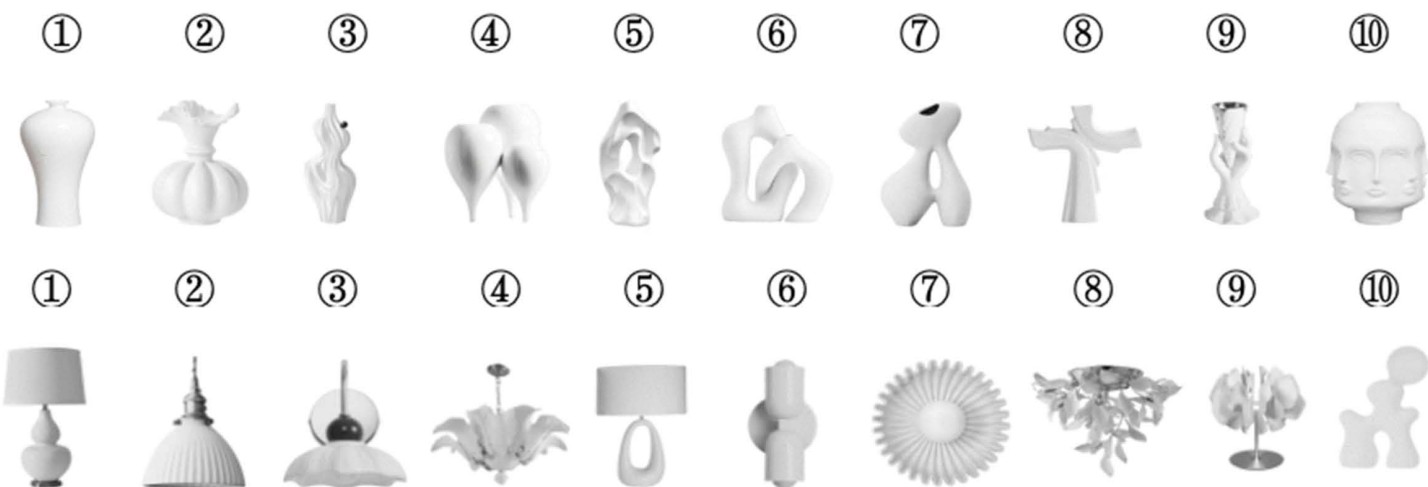

**Fig 1. This figure shows the labels for each ceramic vase (top) and ceramic lighting item (below), used for data analysis.**

adequate sample size, four separate surveys were conducted, each involving 50 participants, for a total of 200 individuals (age range: 18–60 years; mean age: 35 years; SD: 10.5; 51% female). Participants were selected using a simple random sampling method to minimize bias and enhance the sample's representativeness. All participants were native Chinese speakers without a design background, ensuring their evaluations were purely from the consumer perspective, uninfluenced by professional design knowledge. Written informed consent was obtained from each participant prior to conducting the experimental procedures, and data were provided anonymously. Ethical approval for this study was granted by the Ethics Committee of Universiti Putra Malaysia (Reference Number: JKEUPM-2024–304).

## Procedures

The present study collected participants' genuine feedback on product evaluations using a face-to-face survey. Before collecting any data or responses, researchers ensured that all participants provided informed consent and that no personal identifying information was collected during the survey. The Ethics Committee of Universiti Putra Malaysia formally approved the study protocol. After obtaining informed consent, participants were asked to answer three basic screening questions: age, gender, educational level, and professional background.

After completing these demographic questions, participants were instructed to independently evaluate a series of stimuli, specifically 20 product images of 10 ceramic vases and 10 ceramic lighting items. Each stimulus image was accompanied by a set of independent statements, which participants rated using a 7-point Likert scale ranging from "Strongly Disagree" to "Strongly Agree." Aesthetic pleasure is defined as the immediate pleasurable response derived from directly perceiving an object, independent of its functional or utilitarian value [4]. As aesthetic pleasure constitutes the affective basis of aesthetic preference, the present study employed items from the validated Aesthetic Pleasure in Design (APiD) scale to operationalize aesthetic experience. The APiD has been widely applied in subsequent research [38–40], whereas aesthetic preference (AP) scales remain heterogeneous, with multiple versions lacking consensus. To ensure reliability, validity, and comparability, we therefore adopted APiD items rather than AP scales. Moreover, following Rossiter's C-OAR-SE procedure, single-item measures are appropriate when the object is concrete and singular, and the attribute is concrete and easily understood [41]. Bergkvist and Rossiter further demonstrated that for "doubly concrete" constructs, single-item measures perform comparably to multi-item measures in terms of predictive validity [42]. Given that typicality, novelty, and aesthetic preference are concrete, unidimensional, and easily comprehensible constructs, the use of

single-item measures in this study is theoretically justified and consistent with prior empirical work. Accordingly, three key measures were adopted from Blijlevens et al. [4].

Aesthetic preference (aesthetic pleasure): "This vase/lighting is pleasing to see."

Typicality: "This is a typical vase/lighting."

Novelty: "This is a novel vase/lighting."

The study employed four versions of the questionnaire to minimize order effects, utilizing a 2x2 design to control for both stimulus order and statement order. In the first stimulus order group, products were arranged from the most typical to the most novel, whereas the second group used sorting software to generate a random order from 1 to 20. The statement order was also manipulated: in the first group, the order followed the typicality, novelty, and liking sequence, while in the second group, the order was randomly determined by software between 1 and 3. To familiarize participants with the procedure, they were asked to evaluate a product category (cups) that was not part of the main study. The data obtained from this trial evaluation were not included in the analysis. The questionnaire survey data for ceramic vases and lighting items are available in the Supporting Information (S1 and S2 Files).

## Data analysis

This study employed a multi-level statistical analysis strategy to systematically address the proposed research questions. First, a repeated-measures analysis of variance (ANOVA) was conducted to examine differences in preference scores across product design categories and to evaluate the overall effects of typicality and novelty. Subsequently, partial correlation analyses were performed to explore the independent contributions of typicality and novelty to preference formation. Given the repeated-measures nature of the data, generalized estimating equations (GEE) were applied to provide robust parameter estimates for the predictive strength of each factor. Finally, hierarchical regression analysis was used to test whether product category structure moderated the effects of typicality and novelty on aesthetic preference, thereby assessing the boundary conditions of the MAYA design principle across different category types.

## Results

### ANOVA Results

A repeated measures ANOVA was conducted to examine participants' liking ratings for 10 ceramic vases and 10 ceramic lighting items. Design type was the within-subjects factor, while age, gender, and professional background were included as between-subjects variables. The results revealed a significant main effect of design type. Significant differences were observed across designs ($\eta_P^2 vase$ = .071, $p$ < .001; $\eta_P^2 lighting$ = .055, $p$ < .001), indicating that participants' preferences varied meaningfully by design. However, no significant interaction effects were found between liking scores and demographic variables (age, gender, professional background), with all partial eta-squared values being below 0.01. This suggests that demographic factors had a negligible influence on aesthetic preference and were therefore excluded from subsequent regression analyses. Detailed statistical outputs are presented in Table 1 (ceramic vases) and Table 2 (ceramic lighting items).

To further investigate differences in aesthetic evaluation across product categories, this study used repeated-measures ANOVA to analyze participants' preference ratings for 10 ceramic vases and 10 ceramic lighting items. Fig 2 illustrates the mean scores for liking, typicality, and novelty of the ceramic vases. To facilitate comparison, the highest and lowest mean values within each dimension are visually highlighted (bold for the highest and italic for the lowest). Ceramic Vase Stimulus 2 received the highest average liking score (M = 5.35, SD = 1.50); however, it ranked second in typicality (M = 4.23, SD = 1.33) and ninth in novelty (M = 3.60, SD = 1.72). In comparison, Fig 3 illustrates the mean scores for liking, typicality, and novelty of the ceramic lighting items. Using the same visual convention, Ceramic Lighting Stimulus 9 achieved the highest liking score among the lighting products (M = 6.03, SD = 1.06). It ranked fifth in typicality (M = 3.86, SD = 1.54) but first in novelty (M = 5.17, SD = 1.50).

**Table 1. Analysis of Variance (ANOVA) Results for 'Liking' of ceramic vases.**

| | Sum of Squares | $df_{NUM}$ | $df_{DEM}$ | Mean Square | F | p | $\eta^2_P$ |
|---|---|---|---|---|---|---|---|
| Liking | 193.346 | 7.236 | 1251.895 | 26.719 | 13.208 | <.001 | .071 |
| Liking * Age | 32.845 | | | 1.513 | .748 | .790 | .013 |
| Liking * Gender | 12.512 | | | 1.729 | .855 | .545 | .005 |
| Liking *Background | 38.482 | | | 1.773 | .876 | .626 | .015 |
| Liking * Age * Gender * Background | 72.595 | | | 2.006 | .992 | .484 | .028 |

**Table 2. Analysis of Variance (ANOVA) Results for 'Liking' of ceramic lighting items.**

| | Sum of Squares | $df_{NUM}$ | $df_{DEM}$ | Mean Square | F | p | $\eta^2_P$ |
|---|---|---|---|---|---|---|---|
| Liking | 137.627 | 7.224 | 1351.695 | 10.134 | 7.074 | <.001 | .055 |
| Liking * Age | 54.988 | | | 2.537 | 1.350 | .130 | .023 |
| Liking * Gender | 19.517 | | | 2.702 | 1.437 | .184 | .008 |
| Liking *Background | 59.211 | | | 2.732 | 1.453 | .082 | .024 |
| Liking * Age * Gender * Background | 60.879 | | | 2.107 | 1.121 | .301 | .025 |

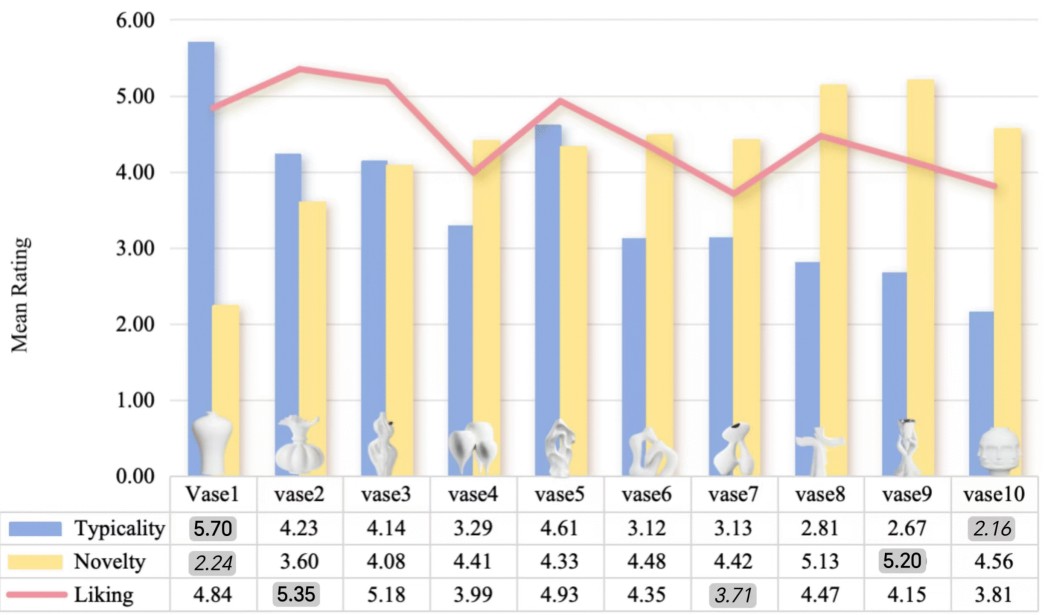

**Fig 2. This figure shows the Mean Score for Liking, Typicality, and Novelty of ceramic vases.** Notes Bold values represent the highest mean scores, while italicized values represent the lowest mean scores.

The ANOVA results, using partial Eta squared ($\eta^2_P$) as a measure of effect size, provided deeper insights into the differences in aesthetic preferences for products from poor categories (ceramic vases) versus rich categories (ceramic lighting items). Tables 3 and 4 present the ANOVA results for both product types on the typicality and novelty scales.

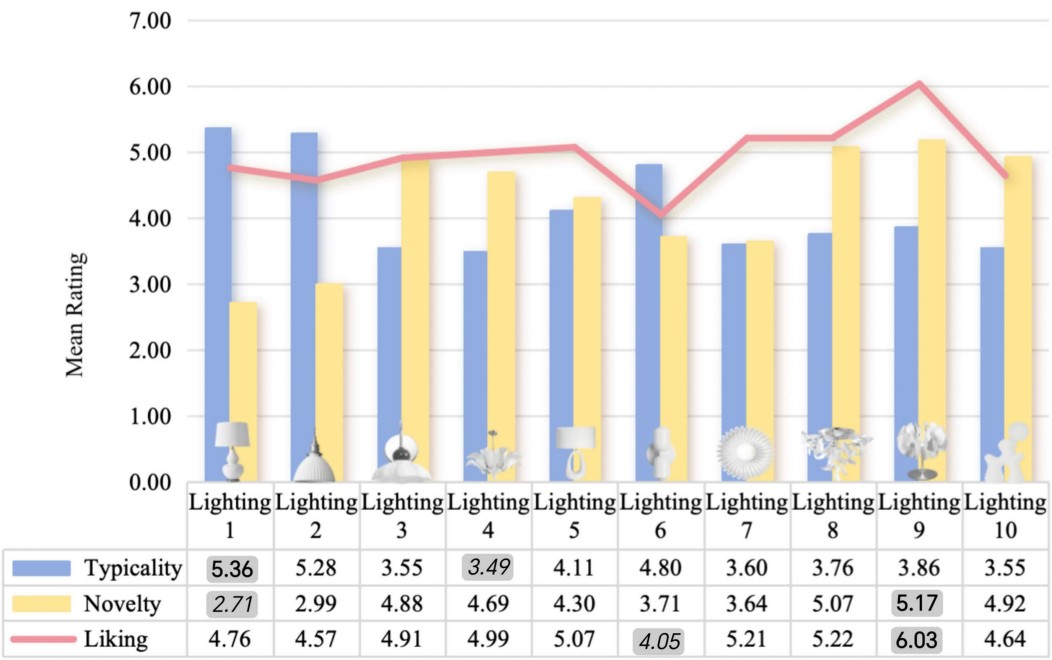

**Fig 3. This figure shows the Mean Score for Liking, Typicality, and Novelty of ceramic lighting items.** Notes Bold values represent the highest mean scores, while italicized values represent the lowest mean scores.

The findings indicate that for ceramic vases, typicality scores were significantly higher than novelty scores ($\eta_P^2$typicality = 0.465, p < 0.001 vs. $\eta_P^2$novelty = 0.304, p < 0.001). In contrast, for ceramic lighting items, novelty scores exceeded typicality scores ($\eta_P^2$novelty = 0.307, p < 0.001 vs. $\eta_P^2$typicality = 0.215, p < 0.001).

### Results of partial correlation test

Two sets of partial correlations were performed for each type of ceramic product (ceramic vases and ceramic lighting items): one controlling for typicality and the other controlling for novelty. This allowed us to isolate and examine each factor's unique contribution to participants' aesthetic preferences while statistically controlling for the others.

Table 5 presents the results of the correlation analysis for ceramic vases and ceramic lighting items. In both categories, typicality was positively associated with aesthetic preference, with a slightly stronger correlation for the ceramic vases (r = .364, p <.001) than for the ceramic lighting items (r = .342, p <.001). Similarly, novelty was positively associated with aesthetic preference in both categories, though the relationship was weaker for the ceramic vases (r = .353, p <.001) than for the ceramic lighting items (r = .364, p <.001). Contrary to expectations, typicality and novelty were negatively correlated (ceramic vases: r = .178; ceramic lighting items: r = .229; both p <.001).

When controlling for typicality, a significant positive partial correlation was observed between novelty and liking in both categories (ceramic vases:r = .314, ceramic lighting items: r = .312; both p <.001), indicating that novelty remained significantly associated with aesthetic preference even after accounting for the effect of typicality. This finding further supports the independent contribution of novelty to shaping aesthetic preferences.

When controlling for novelty, a significant positive partial correlation was found between typicality and liking in both categories (ceramic vases:r = .327, ceramic lighting items: r = .286; both p <.001), indicating that typicality remained significantly associated with aesthetic preference even after accounting for the influence of novelty. This result further supports the independent contribution of typicality to aesthetic preference.

**Table 3. Analysis of Variance (ANOVA) Results of all scales for Ceramic Vases.**

| | df$_{NUM}$ | df$_{DEM}$ | Epsilon | F | P | $\eta^2_P$ |
|---|---|---|---|---|---|---|
| **Liking** | 7.272 | 1447.184 | 0.842 | 41.664 | .001 | .173 |
| Typicality | 7.050 | 1402.895 | 0.815 | 173.139 | .001 | .465 |
| Novelty | 6.925 | 1378.109 | 0.800 | 86.718 | <.001 | .304 |

**Table 4. Analysis of Variance (ANOVA) Results of all scales for Ceramic Lighting items.**

| | df$_{NUM}$ | df$_{DEM}$ | Epsilon | F | P | $\eta^2_P$ |
|---|---|---|---|---|---|---|
| **Liking** | 7.155 | 1423.852 | 0.828 | 34.153 | .001 | .146 |
| Typicality | 7.587 | 1509.864 | 0.880 | 54.509 | .001 | .215 |
| Novelty | 7.456 | 1483.696 | 0.864 | 88.352 | <.001 | .307 |

**Table 5. Results of Partial Correlation Test for Ceramic Vases(typicality as Controlled Variable).**

| Control Variable | Variable Pair | Category | r | p |
|---|---|---|---|---|
| −none−[a] | Liking-Typicality | Vases | .364 | <.001 |
| | | Lighting | .342 | <.001 |
| | Liking-Novelty | Vases | .353 | <.001 |
| | | Lighting | .364 | <.001 |
| | Typicality-Novelty | Vases | .178 | <.001 |
| | | Lighting | .229 | <.001 |
| Typicality | Liking-Novelty | Vases | .314 | <.001 |
| | | Lighting | .312 | <.001 |
| Novelty | Liking-Typicality | Vases | .327 | <.001 |
| | | Lighting | .286 | <.001 |

Table notes: Cells contain zero-order (Pearson) correlations and p-values are unadjusted for repeated measures.

## Results of the Generalized Estimating Equation (GEE) analysis

This study further employed the Generalized Estimating Equations (GEE) method to evaluate the predictive strength of each independent variable for the dependent variable, aesthetic preference. Typicality and novelty were included as independent variables to assess their influence on design preference further. The resulting $\beta$ coefficients (B values) provided additional validation of the effect sizes previously calculated using partial Eta squared. Tables 6 and 7 present the GEE analysis results for both ceramic vases and ceramic lighting items on the "Liking" variable.

The findings revealed that for ceramic vases, typicality had a more substantial influence on preference ratings than novelty (Btypicality = 0.198, p < .001; Bnovelty = 0.086, p < .001). In contrast, forceramic lighting items, novelty exerted a greater influence than typicality (Bnovelty = 0.119, p < .001; Btypicality = 0.069, p < .001).

## Results of the regression analysis

Finally, a hierarchical regression was conducted with liking as the dependent variable and typicality and novelty as predictors. In the first step, typicality and novelty were entered; in the second step, product category (1 = ceramic vases, 2 = ceramic lighting items) and their interaction terms were added. The second model showed a significant change in $R^2$ ($\triangle R^2$ = .022), indicating that product category moderates the effect of typicality and novelty on aesthetic preference

 

**Table 6. Summary of Generalized Estimating Equation Analysis for Variables Predicting Pleasing to See for Ceramic Vases.**

| Variable | B | SE B | 95% CI for B | P |
|---|---|---|---|---|
| **Typicality** | .198 | .0220 | [0.155,0.241] | .001 |
| **Novelty** | .086 | .0201 | [0.046,0.125] | <.001 |

Table notes B indicates Unstandardized Beta, SE B indicates Standard Error for the Unstandardized Beta, CI indicates Confidence Interval. (N = 2000).

**Table 7. Summary of Generalized Estimating Equation Analysis for Variables Predicting Pleasing to See for Ceramic Lighting items.**

| Variable | B | SE B | 95% CI for B | P |
|---|---|---|---|---|
| **Typicality** | .069 | .0221 | [0.029,0.108] | .001 |
| **Novelty** | .119 | .0196 | [0.071,0.167] | <.001 |

Table notes B indicates Unstandardized Beta, SE B indicates Standard Error for the Unstandardized Beta, CI indicates Confidence Interval. (N = 2000)

($R^2 = .126$, $F(5, 3994) = 114.946$, $p < .001$), confirming that product type moderates their influence. Typicality ($\beta = -.166$, $p < .001$) and novelty ($\beta = .289$, $p < .001$) both predicted liking. The interactions were significant: $\beta$ typicality $\times$ product $= .505$ ($p < .001$), $\beta$ novelty $\times$ product $= -.131$ ($p = .030$).

For ceramic vases, typicality and novelty explained 21.8% of the variance ($R^2 = .218$, $F(2, 197) = 27.416$, $p < .001$), with typicality ($\beta = .311$) slightly more influential than novelty ($\beta = .297$). For ceramic lighting items, the pattern was reversed but balanced: typicality ($\beta = .273$) and novelty ($\beta = .302$) jointly explained 20.3% of variance ($R^2 = .203$, $F(2, 197) = 25.144$, $p < .001$).

## Discussion

This study provides a systematic investigation into how typicality and novelty influence consumer aesthetic preferences, with a particular focus on whether product category structure (poor vs. rich categories)moderates this relationship. A multi-phase data analysis approach was employed, including repeated-measures analysis of variance (ANOVA), partial correlation analysis, generalized estimating equations (GEE), and hierarchical regression. The findings reveal several important theoretical insights:

First, although most previous studies have reported a negative correlation between typicality and novelty, recent empirical studies suggest that their relationship may vary across different design domains. For example, Ceballos et al. examined aesthetic preferences for apparel products and found a positive correlation between typicality and novelty [1]. Similarly, Maluleem investigated Thai souvenirs and reported a positive relationship between the two constructs [42]. In the ceramic products examined in the present study, a similar positive correlation was observed. For instance, ceramic lighting designs retained the prototypical structure of a lamp (high typicality) while using ceramic materials uncommon in lighting design (high novelty). The use of such innovative materials enhanced the perceived novelty without disrupting prototype recognition, allowing typicality and novelty to coexist and even correlate positively.

Second, in the poor category products (represented by ceramic vases), typicality was found to be the primary determinant of consumer aesthetic preference. Specifically, GEE results showed that the predictive effect of typicality (B = 0.198) was significantly more substantial than that of novelty (B = 0.086). Partial correlation analysis further confirmed that, even after controlling for novelty, typicality remained a stronger, statistically significant predictor of preference. This finding aligns with prior research suggesting that poor category products are characterized by high typicality and low novelty [33,34,37,38,43]. The inconsistent results reported by Suhaimi et al. for poor categories may be attributed to their use of industrial boilers as stimuli—products that are largely utilitarian and not typically evaluated based on visual appearance

[44]. Overall, these results support the theory of processing fluency, which posits that designs consistent with familiar category schemas reduce cognitive load and enhance aesthetic pleasure [45,46]. In poor categories, emotional responses tend to be strongest for the most prototypical exemplars—that is, for items that most closely conform to category expectations [30]. Such stimuli require minimal information processing, thereby facilitating recognition and eliciting a positive affective response [31,32].

Third, in the rich category products(represented by ceramic lighting), novelty emerged as the more salient factor in attracting consumer attention. GEE analysis indicated that novelty (B = 0.119) had a greater predictive effect than typicality (B = 0.069). Partial correlations also demonstrated that novelty continued to significantly predict aesthetic preference even after controlling for typicality. This finding is consistent with previous studies suggesting that rich categories are generally characterized by low typicality and high novelty [33,34]. Theoretically, this suggests that consumers in rich product categories are more receptive to novel designs, as these categories allow for greater variation and creativity. Novel designs provide stronger perceptual stimulation and exploratory appeal, making them key drivers of aesthetic value in such contexts.

Fourth, this study confirmed that product category structure significantly moderates the effect of typicality and novelty on aesthetic preference, thereby refining the boundary conditions of the MAYA (Most Advanced Yet Acceptable) design principle. Hierarchical regression analysis revealed significant interaction effects (typicality $\times$ category $\beta$ =0.505, p < 0.001; novelty $\times$ category $\beta$ = −0.131, p = 0.030), indicating that the balance between novelty and typicality shifts depending on the poor and rich of the product category. This finding enhances the theoretical framework of the MAYA principle by highlighting the importance of contextual factors such as product category structure.

Furthermore, regression results showed that typicality and novelty together accounted for 21.8% of the variance in aesthetic preference for ceramic vases ($R^2$ = 0.218), and 20.3% for ceramic lighting items ($R^2$ = 0.203). These results provide quantitative evidence that both factors are critical in shaping visual aesthetic responses to product design.

Whereas previous studies have emphasized the dynamic balance between novelty and typicality in aesthetic judgment [23,27], few have systematically considered structural differences between product categories. This study empirically demonstrates that product category structure moderates the applicability of the MAYA principle. Taken together, these findings provide clear empirical support for the proposed hypothesis, showing that rich product categories exhibit greater tolerance for novelty, whereas poor categories favor higher typicality. These findings are consistent with prior research [33,34]. While earlier studies primarily focused on furniture products, the current research extends this understanding to ceramic design, thereby broadening the applicability of category-based aesthetic theory. These findings not only support the theoretical balance proposed by the MAYA principle but also clarify how that balance may shift depending on a product's structural nature. By doing so, this research contributes to a more nuanced and context-sensitive application of the MAYA framework, offering a clear theoretical foundation for future empirical studies.

## Conclusion

This study reveals how product category structure significantly influences the mechanisms underlying consumer aesthetic preferences, clarifying the relative importance of typicality and novelty across different product types. In line with the central hypothesis guiding this study, the overall pattern of results consistently demonstrates that the relative roles of typicality and novelty in aesthetic preference depend on product category structure. By empirically examining both poor and rich categories within the domain of ceramic design, this study not only extends category-based aesthetic theory but also confirms the applicability of the 'Most Advanced Yet Acceptable' (MAYA) principle across different category structures. Specifically, the results reveal that typicality and novelty interact to predict aesthetic preference, validating the MAYA framework's core assumption that optimal design outcomes emerge from a balance between familiarity and innovation.

Theoretically, this research advances the understanding of aesthetic preference formation by identifying product category structure as a key moderating factor influencing the perceived acceptability of novel designs. From a practical perspective, designers should tailor their aesthetic strategies according to category structure: in poor categories, emphasizing

high typicality and low novelty enhances consumer acceptance and perceived coherence, whereas in rich categories, highlighting high novelty and low typicality fosters aesthetic interest and differentiation.

Most importantly, this study's findings support the proposed hypothesis that rich product categories are more tolerant of novelty than poor ones. Specifically, typicality played a dominant role in shaping aesthetic preference for ceramic vases (poor categories), whereas novelty exerted a stronger influence in ceramic lighting products (rich categories). This conclusion completes the empirical test of the hypothesis and clarifies the boundary conditions under which the MAYA principle operates.

Despite these contributions, two primary limitations should be acknowledged. First, all participants in this study were recruited from China, which may restrict the cultural generalizability of the findings. Future research could expand the sample to include diverse cultural contexts, particularly through cross-national comparisons between Chinese and European consumers, to test the cross-cultural robustness of the results. Second, this study focused exclusively on visual aesthetics, neglecting the potential influence of other sensory dimensions such as auditory or tactile perception. Future research should consider multi-sensory approaches to aesthetic evaluation to better understand how product design holistically shapes user experience.

In summary, by integrating typicality, novelty, and category structure into the aesthetic evaluation process, this study offers a more nuanced and context-sensitive interpretation of the MAYA principle and lays a solid foundation for future empirical research on product aesthetics.

## Supporting information

**S1 File. Questionnaire survey data of ceramic vases.**
(XLSX)

**S2 File. Questionnaire survey data of ceramic lighting items.**
(XLSX)

**S1 Checklist. Inclusivity in global research.**
(PDF)

## Acknowledgments

This study is part of the first author's Ph.D. thesis at the Faculty of Design and Architecture, Universiti Putra Malaysia. The authors would like to express their gratitude to Professor Allan Whitfield, Dr. Deirdre Barron, and Dr. Lichen Tai for their assistance with this study.

## Author contributions

**Conceptualization:** Qi Wu.

**Data curation:** Qi Wu, Qianhui Ren.

**Formal analysis:** Qi Wu.

**Investigation:** Qi Wu.

**Methodology:** Qi Wu.

**Writing – original draft:** Qi Wu.

**Writing – review & editing:** Qi Wu, Mohd Faiz bin Yahaya, Lichen Tai.

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
