## [Decision Letter · Decision Letter 0]

1 Oct 2025

PONE-D-25-32602Practical research on the boundaries of MAYA design Principles with Ceramic Products as the CarrierPLOS ONE

Dear Dr. Wu,

Thank you for submitting your manuscript to PLOS ONE. After careful consideration, we feel that it has merit but does not fully meet PLOS ONE’s publication criteria as it currently stands. Therefore, we invite you to submit a revised version of the manuscript that addresses the points raised during the review process.

Please focus on everything posted by reviewers and resubmit

We look forward to receiving your revised manuscript.

Kind regards,

Carlos P. Odriozola, Ph.D

Academic Editor

PLOS ONE

Journal Requirements:

Reviewers' comments:

Reviewer's Responses to Questions

**Comments to the Author**

1. Is the manuscript technically sound, and do the data support the conclusions?

Reviewer #1: Yes

Reviewer #2: Yes

2. Has the statistical analysis been performed appropriately and rigorously? 

Reviewer #1: Yes

Reviewer #2: Yes

3. Have the authors made all data underlying the findings in their manuscript fully available?

Reviewer #1: No

Reviewer #2: Yes

4. Is the manuscript presented in an intelligible fashion and written in standard English?

Reviewer #1: Yes

Reviewer #2: Yes

5. Review Comments to the Author

Reviewer #1: This exquisitely written paper examines the MAYA principle in ceramic products by measuring the effects of the aesthetic properties of typicality and novelty on aesthetic preference. The study focuses on ceramic products, specifically two distinct categories: open and closed. The authors included 10 stimuli for the open category and 10 for the closed, selecting vases and lighting objects to represent both categories. A within-subjects experimental design was implemented to test the objectives using Chinese participants. Measures were taken to avoid order effects on both the stimuli and the measures. The study demonstrates that the MAYA principle applies to both categories and further confirms the moderating role of category type.

Great work.

The suggestions in the enclosed document are intended to help improve the rigor and clarity of your paper.

Best of luck with the revision!

I really enjoyed the study, even though it includes only one experiment. Below, I have included some suggestions to help make the paper more rigorous and clear for readers. I believe it is an excellent study that deserves publication. Many of my suggestions are cosmetic, but some address justification and clarification. Because this is a study in the social sciences in PLOS ONE—even though it is focused on product design and consumer behavior—I am convinced that the inclusion of hypotheses is necessary. That is the major change I request. So, it is small major revision in my opinion.

In Summary, these are the main changes I believe will improve the manuscript:

- Clarify writing and improve consistency: Revise unclear sentences (e.g., in the abstract) and maintain consistent order when presenting key concepts like typicality vs. novelty and vases vs. lighting.

- Strengthen methodological transparency: Add details about the expert panel, stimulus selection, and measurement procedures, including justifications for construct-scales choices and sources.

- Enhance rigor and scientific contribution: Include formal hypotheses, verify construct definitions and usage (e.g., unity or typicality? Aesthetic pleasure or aesthetic preference?), and support theoretical claims in the discussion with relevant citations.

- Improve presentation of results and key findings: Make figures and tables easier to interpret by adding labels, totals, and clarifications; visually highlight key values. Explicitly state in the conclusion that the study confirms the MAYA principle across both product categories.

Reviewer #2: This interesting manuscript investigates the MAYA principle through the interplay of typicality and novelty drives preference across object domains with differing categories. Specifically, it applies and extends the MAYA principle to examine how typicality and novelty predict aesthetic preferences for ceramic products across open versus closed object categories, using survey data from 200 participants. The authors report that typicality drives preferences in closed categories, while novelty is more influential in open categories. Overall, the paper is clear and well-motivated and addresses an interesting and important question in design aesthetics.

However, there are still some concerns (concepts, stimulus-selection procedure, and positive correlation result) that must be addressed to strengthen the paper’s clarity and rigor.

1. Concept

On p. 3, line 70, the authors invoke Whitfield’s (2005) model (named Categorical-Motivational model, the CM model), which states that object category influences the weight of typicality and novelty in shaping aesthetic preference.

As I understand, the CM model includes some types of categories, with open and closed categories at two extremes, partially open categories between these two extremes, and rich and poor categories within the partially open categories (see some early work from Whitfield, 1983, 2000, 2005, 2009; Whitfield & Slatter, 1979; and some empirical evidence, e.g., Suhaimi, 2023; Chen et al., 2025). There is something unclear about why the authors defined vases are open categories and lightings are closed categories. Thus, I would ask the authors to clarify the reasons. This may help the reader to understand better.

NB: To my understanding, I would suggest the stimuli used in this manuscript (vase and lighting) are defined as rich and poor categories rather than open and closed categories.

2. Stimulus-selection procedure

As the study aims to compare the typicality and novelty in shaping aesthetic preference across two types of products. It is necessary to select appropriate stimuli to control the independent variables, typicality and novelty, at a comparable level. In more direct words, you cannot select ten typical products versus ten novel products. I would ask the authors to clarify the procedure for how the two types of products are at a comparable level in typicality and novelty.

3. Positive correlation between typicality and novelty

It is an interesting result that shows a positive correlation between typicality and novelty, different from some previous studies and our cognition. This is worth exploring and potentially reporting, as it could strengthen the interpretation.

NB: I would suggest that the authors can discuss it from the definitions of typicality and novelty, and the characterizations of the stimuli.

Minor points:

- Tables 7, 8, 9 & 10: Combine them into a single table that directly shows the correlation result.

- Create a figure to directly show typicality, novelty, and liking to replace figures 2&3.

- Report the demographic and reliability results before data analysis.

Reference:

Whitfield, T. W. A. (1983). Predicting preference for familiar, everyday objects: An experimental confrontation between two theories of aesthetic behaviour. Journal of Environmental Psychology, 3(3), 221–237. https://doi.org/10.1016/S0272-4944(83)80002-4

Whitfield, T. W. A. “Beyond Prototypicality: Toward a Categorical-Motivation Model of Aesthetics.” Empirical Studies of the Arts 18, no. 1 (January 2000): 1–11. https://doi.org/10.2190/KM3A-G1NV-Y5ER-MR2V.

Whitfield, T. W. A. (2005). Aesthetics as Pre-linguistic Knowledge: A Psychological Perspective. Design Issues, 21(1), 3–17. https://doi.org/10.1162/0747936053103002

Whitfield, T. W. Allan. “Theory Confrontation: Testing the Categorical-Motivation Model.” Empirical Studies of the Arts 27, no. 1 (January 2009): 43–59. https://doi.org/10.2190/EM.27.1.c.

Whitfield, T. W. A., & Slatter, P. E. (1979). The effects of categorization and prototypicality on aesthetic choice in a furniture selection task. British Journal of Psychology, 70(1), 65–75. https://doi.org/10.1111/j.2044-8295.1979.tb02144.x

Suhaimi, S. N., Kuys, B., Barron, D., Li, N., Rahman, Z., & Whitfield, A. (2023). Probing the Extremes of Aesthetics: The Role of Typicality and Novelty in the Aesthetic Preference of Industrial Boilers. Empirical Studies of the Arts, 41(1), 216–230. https://doi.org/10.1177/02762374221094137

Chen, S., Whitfield, A., Barron, D., Zahari, Z. A., Suhaimi, S. N., Huang, L., & Wang, Y. (2025). Categorization and Aesthetic Preference: Examining Typicality and Novelty Across Rich and Poor Categories. Empirical Studies of the Arts, https://doi.org/10.1177/02762374251371282

6. PLOS authors have the option to publish the peer review history of their article (what does this mean? ). If published, this will include your full peer review and any attached files.). If published, this will include your full peer review and any attached files.

**Do you want your identity to be public for this peer review?** For information about this choice, including consent withdrawal, please see our For information about this choice, including consent withdrawal, please see our Privacy Policy ..

Reviewer #1: No

Reviewer #2: No

While revising your submission, please upload your figure files to the Preflight Analysis and Conversion Engine (PACE) digital diagnostic tool, https://pacev2.apexcovantage.com/ . PACE helps ensure that figures meet PLOS requirements. To use PACE, you must first register as a user. Registration is free. Then, login and navigate to the UPLOAD tab, where you will find detailed instructions on how to use the tool. If you encounter any issues or have any questions when using PACE, please email PLOS at . PACE helps ensure that figures meet PLOS requirements. To use PACE, you must first register as a user. Registration is free. Then, login and navigate to the UPLOAD tab, where you will find detailed instructions on how to use the tool. If you encounter any issues or have any questions when using PACE, please email PLOS at figures@plos.org . Please note that Supporting Information files do not need this step.. Please note that Supporting Information files do not need this step.

---

## [Author Response · Author response to Decision Letter 1]

3 Nov 2025

We sincerely thank the editor and reviewers for their valuable comments on our manuscript. The editor and Reviewer 1 provided four specific suggestions and offered detailed guidance for revisions in each section of the paper, which has been extremely helpful. Reviewer 2 provided six comments along with targeted suggestions, which not only helped us clarify the core concepts of the manuscript more accurately but also improved the presentation of figures and tables. We have carefully considered all the feedback and have made corresponding revisions throughout the manuscript.

---

## [Decision Letter · Decision Letter 1]

14 Jan 2026

PONE-D-25-32602R1Practical research on the boundaries of MAYA design Principles with Ceramic Products as the CarrierPLOS One

Dear Dr. Yahaya,

Thank you for submitting your manuscript to PLOS ONE. After careful consideration, we feel that it has merit but does not fully meet PLOS ONE’s publication criteria as it currently stands. Therefore, we invite you to submit a revised version of the manuscript that addresses the points raised during the review process.

We look forward to receiving your revised manuscript.

Kind regards,

Annesha Sil, Ph.D.

Staff Editor

PLOS One

Journal Requirements:

Reviewers' comments:

Reviewer's Responses to Questions

**Comments to the Author**

1. If the authors have adequately addressed your comments raised in a previous round of review and you feel that this manuscript is now acceptable for publication, you may indicate that here to bypass the “Comments to the Author” section, enter your conflict of interest statement in the “Confidential to Editor” section, and submit your "Accept" recommendation.

Reviewer #1: All comments have been addressed

Reviewer #2: All comments have been addressed

2. Is the manuscript technically sound, and do the data support the conclusions?

Reviewer #1: Yes

Reviewer #2: Yes

3. Has the statistical analysis been performed appropriately and rigorously? 

Reviewer #1: Yes

Reviewer #2: Yes

4. Have the authors made all data underlying the findings in their manuscript fully available?

Reviewer #1: Yes

Reviewer #2: Yes

5. Is the manuscript presented in an intelligible fashion and written in standard English?

Reviewer #1: Yes

Reviewer #2: Yes

6. Review Comments to the Author

Reviewer #1: Review PONE-D-25-32602R1

Review sent on Jan 7, 2026

Thank you to the authors for the great work on the revision. I have only two very minor suggestions before publication.

1)I note that the manuscript formulates a single hypothesis. While this is not common, it is acceptable for PLOS ONE, given the journal’s emphasis on methodological rigor rather than the number of hypotheses. However, the current presentation does not sufficiently emphasize this hypothesis as the central organizing element of the study.

With minor revisions, I recommend making the hypothesis more visible and clearly articulated throughout the manuscript. The Conclusions should then explicitly state whether the hypothesis is supported or not, as this section should tie together the study’s objectives, analyses, and main contributions. While a similar point is addressed in the final part of the Discussion, I believe it should be reiterated in the Conclusions to clearly close the loop of the hypothesis testing.

2) Tables and Figures 2 and 3 are now fantastic and easier to understand! Great way to unify and make it comparable and contrasting. When explaining figures in which the highest and lowest values are highlighted or italicized, please include a note explaining these markings, either in the text or, ideally, in a note below each specific figure. These markings may not be obvious to a reader.

Reviewer #2: The main concerns have been formally addressed in the 2nd manuscript. I do not see the need for another round of revisions.

In detail, the author has changed the definition from "open-closed" to "rich-poor" categories, and I agree with that. For these two kinds of objects (vase and lighting), "rich-poor" categories are more suitable for our knowledge. Besides, the author has detailed the stimulus selection procedure and improved the experimental transparency; my concerns have been addressed. After that, the author discussed the positive correlation between typicality and novelty and gave explanation for that. Lastly, minor points have been revised.

So I agree to accept this current version. Merry Christmas.

7. PLOS authors have the option to publish the peer review history of their article (what does this mean? ). If published, this will include your full peer review and any attached files.). If published, this will include your full peer review and any attached files.

**Do you want your identity to be public for this peer review?** For information about this choice, including consent withdrawal, please see our For information about this choice, including consent withdrawal, please see our Privacy Policy ..

Reviewer #1: **Yes:** Lina M. CeballosLina M. Ceballos

Reviewer #2: No

---

## [Author Response · Author response to Decision Letter 2]

25 Jan 2026

We sincerely appreciate the editor and all reviewers for their meticulous evaluation, insightful comments and constructive suggestions on our manuscript. These valuable opinions have greatly helped us to identify the deficiencies of the study and improve the quality of the manuscript comprehensively.

---

## [Editor Report · Decision Letter 2]

29 Jan 2026

Practical research on the boundaries of MAYA design Principles with Ceramic Products as the Carrier

PONE-D-25-32602R2

Dear Dr. Yahaya,

We’re pleased to inform you that your manuscript has been judged scientifically suitable for publication and will be formally accepted for publication once it meets all outstanding technical requirements.

Kind regards,

Annesha Sil, Ph.D.

Staff Editor

PLOS One
---

## [Editor Report · Acceptance letter]

PONE-D-25-32602R2

PLOS One

Dear Dr. Yahaya,

I'm pleased to inform you that your manuscript has been deemed suitable for publication in PLOS One. Congratulations! Your manuscript is now being handed over to our production team.

Kind regards,

on behalf of

Dr Annesha Sil

Staff Editor

PLOS One